# Performance Analysis of Ku/Ka Dual-Band SAR Altimeter from an Airborne Experiment over South China Sea

**Xiaonan Liu** [1,2] , **Weiya Kong** [1], **Hanwei Sun** [1,\*] **and Yaobing Lu** [1]

1    Beijing Institute of Radio Measurement, Beijing 100854, China; liuxn0816@htbirm.cn (X.L.);
kongweiya@buaa.edu.cn (W.K.); luyaobing65@htbirm.cn (Y.L.)
2    The Graduate School of Second Academy of China Aerospace, Beijing 100854, China
\*    Correspondence: sunhw12@tsinghua.org.cn; Tel.: +86-13488682636

**Abstract:** Satellite radar altimeters have been successfully used for sea surface height (SSH) measurement for decades, gaining great insight in oceanography, meteorology, marine geology, etc. To further improve the observation precision and spatial resolution, radar altimeters have evolved from real aperture to synthetic aperture, from the Ku-band to Ka-band. Future synthetic aperture radar (SAR) altimeter of the Ka-band is expected to achieve better performance than its predecessors. To verify the SAR altimeter data processing method and explore the system advantage of the Ka-band, a Ku/Ka dual-band SAR altimeter airborne experiment was carried out over South China Sea on 6 November 2021. Through dedicated hardware design, this campaign has acquired the Ku and Ka dual-band echo data simultaneously. The airborne data are processed to estimate the SSH retrieval precision after a series of procedures (including height compensation, range migration correction, multi-look processing, waveform re-tracking). To accustom to the airborne experiment design, a SAR echo model that fully considers both the attitude variation of the aircraft and the elliptical footprint of radar beam is established. The retrieved SSH data are compared with the public SSH data along the flight path at the experiment day, showing good consistence for both bands. By calculating the theoretical precision of waveform re-tracking and re-processing the dual-band airborne data into different bandwidths, it is demonstrated that the Ku/Ka precision ratio is possible to achieve 1.4 within the 27 km offshore area, which indicates that Ka-band has better performance.

**Keywords:** Ka-band; SAR altimeter; airborne experiment; waveform re-tracking

## 1. Introduction

Since the first space-borne radar altimeter on geodetic satellite (GEOSAT) acquired high precision sea surface height (SSH) data, satellite radar altimeter has been utilized to monitor the ocean circulation on a global scale [1]. The Topex/Poseidon satellite and Jason series satellites have observed the ocean surface topography for decades and provided marine remote sensing data of high quality [2,3]. By re-tracking the sea surface echoes, the radar altimeter can obtain key parameters such as sea surface height, significant wave height (SWH), backscatter coefficient, and wind speed. These parameters can be applied to geophysics, ocean dynamics, atmospheric environment, ice detection, etc. [4].

With the growing demand for higher resolution and precision, conventional altimeter can no longer meet the requirements limited by its limited antenna size which works in real aperture mode [5,6]. Therefore, the idea of synthetic aperture radar (SAR) altimeter has been developed and become the new generation of satellite altimeter [7]. The Cryosat-2 satellite which was launched in April 2010, has successfully tested the synthetic aperture mode for the first time [8]. The SAR radar altimeter (SRAL) on Sentinel-3A satellite, which was launched in February 2016, is operating in SAR mode on a global scale, and has achieved better precision than previous altimeters [9]. The latest Poseidon-4 altimeter on Jason-CS (or Sentinel-6 MF), which was launched on 21 November 2020, has adopted a

new "interleaved mode", which is expected to gain higher resolution and measurement precision [10].

All of the above-mentioned altimeters operate in Ku-band for ranging (C-band is mainly used for ionospheric correction), yet benefitting from the lower noise level and higher spatial resolution, Ka-band altimeter is attracting growing attention. Moreover, compared with Ku-band, Ka-band altimeter does not need the extra C-band for minor ionospheric delay, which simplifies the system design [11]. The first Ka-band altimeter ALtika on SARAL satellite, has proved to have better performance than the Ku/C altimeter of Jason-2 [12,13]. The Surface Water and Ocean Topography (SWOT) mission, which is scheduled to launch in 2022, has as its primary instrument a Ka-band radar interferometer (KaRIn). The KaRIn is believed to be the next generation of satellite altimeter, since it works closer to an interferometric SAR that can measure two-dimensional SSH over a wide swath, rather than a nadir looking altimeter. This mission is a collaboration between the National Aeronautics and Space Administration (NASA) and Centre National d'Etudes Spatiales (CNES), and is planned to achieve 3 cm precision on a 1 km grid over its 120 km wide swath [14].

Through the above analysis, it can be concluded that Ka-band SAR altimeter has great potential and broad prospect for future development. China's new "Guanlan" science satellites, will equip dual-frequency (Ku-band and Ka-band) SAR altimeters for the first time to advance oceanic remote sensing [15]. The Beijing Institute of Radio Measurement has tested the data processing method of Ku-band and Ka-band SAR altimeters by conducting a series of airborne experiment. The first airborne campaign took place in 2019 over the Qingdao Coast Sea, China, and the second one was conducted in 2020 over Rizhao Coast Sea, China. Both experiments obtained satisfying results, which proved the feasibility of Ka-band altimeter [16]. To further verify the SAR altimeter data processing method and explore the system advantage of Ka-band, a Ku/Ka dual-band altimeter airborne experiment was carried out over the South China Sea on 6 November 2021. The airborne data of Ku and Ka-band are processed to compare the SSH retrieval precision after a series of procedures.

This paper is based on the 2021 airborne campaign. The airborne instrument's primary mode is interferometric altimetry, thus both frequencies are equipped with rectangular dual-channel antennas. During the experiment, the beam is adjusted to point at nadir, therefore the nadir altimeter echoes are obtained, which are the processing object of this paper. The antenna footprint is not a round shape which is usually the case for satellite altimeters, besides, the aircraft attitude changes dramatically. Thus, the echo of this airborne experiment has some particularities and needs to be considered during the whole data processing.

In Section 2, after introducing some basic information of the airborne experiment, the critical steps of the SSH determination principle of the airborne SAR altimeter (including echo model establishment, re-tracking algorithm and theoretical precision analysis) are elaborated. A SAR echo model that fully considers the attitude change of the aircraft and is applicable to elliptical radar beam is established firstly, then the method for evaluating the theoretical re-tracking precision is illustrated, which plays a significant role in dual-band airborne data re-processing.

In Section 3, the raw dual-band airborne echoes are performed synthetic aperture processing, and the obtained multi-look SAR waveforms are re-tracked. The SSH retrieval results of Ku and Ka-band are compared with the public SSH data along the flight path at the experiment day [17]. The results indicate that Ka-band has no obvious ranging advantage in comparison to Ku-band. By analyzing the theoretical precision from pre-established method and conducting low-pass filtering to evaluate the ranging precision of different signal bandwidth, it is shown that Ka-band should have higher precision if a larger bandwidth is adopted. The re-analyzed results demonstrate that the Ku/Ka precision ratio is possible to achieve 1.4 within the 27 km offshore area by taking into account the different expected bandwidths on satellite-borne Ku and Ka band altimeters.

Discussion and conclusions are in Sections 4 and 5.

## 2. Materials and Methods

### 2.1. The Airborne SAR Altimetry Experiment

The main purpose of conducting nadir altimetry in the airborne experiment is to verify the data processing method of SAR altimeter. To compare the performance of Ku and Ka-band, the airborne SAR altimeter system (ASAS) is specially designed to work in Ku/Ka dual-band simultaneously. This configuration makes the data of two bands completely comparable.

The airborne campaign took place on 6 November 2021. A Cessna-208B aircraft was used to carry the ASAS instrument together with the airborne Global Navigation Satellite System (GNSS) and the inertial navigation unit. The ASAS hardware mainly includes the antenna system and electronic equipment cabin. Figure 1 shows the details of Ka/Ku dual-band antennas mounted on a high precision servo which should maintain the attitude to required precision. Both of the transmitting antennas are rectangular, which gives the radar beam an elliptical shape. Since each band has two receiving antennas, the altimeter system can also operating in interferometric SAR mode.

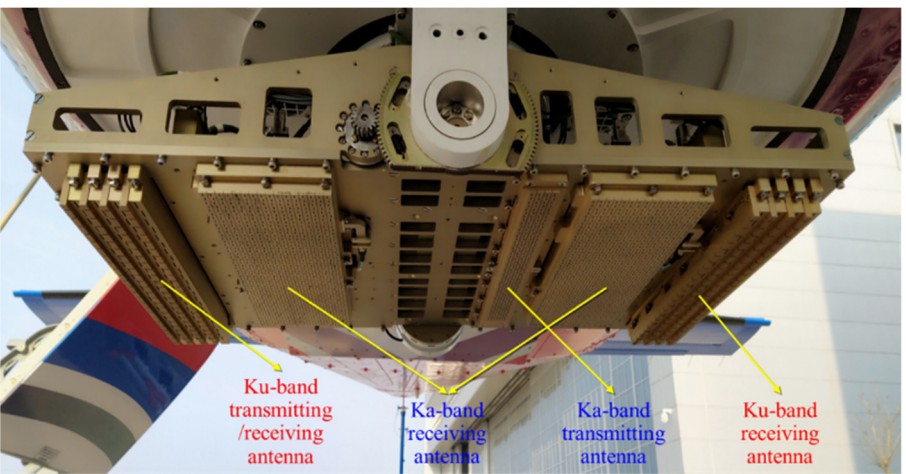

**Figure 1.** The antennas of the airborne SAR altimeter system.

In order to enlarge the across-range swath as much as possible, the mean flight altitude was about 3767 m. Depending on the antenna aperture and flight altitude, the along-track footprint of the 3 dB beam of Ku-band and Ka-band are about 180 m and 90 m, respectively. Although the across-track footprint is larger, it is not larger than 1 km. This relative narrow footprint makes the echo waveform easily distorted by speckle noise originated from sea surface wave, so some extra processing is needed to be performed to improve the signal-to-clutter ratio (SCR). The main parameters of the ASAS are listed in Table 1.

**Table 1.** The main parameters of the airborne SAR altimeter system.

| Parameter | Ku-Band | Ka-Band |
|---|---|---|
| Carrier frequency | 15.8 GHz | 35.8 GHz |
| Along-track beam width (3 dB) | 2.82° | 1.36° |
| Across-track beam width (3 dB) | 15° | |
| Pulse Width | 15 µs | |
| Bandwidth | 900 MHz | |
| Pulse repetition frequency | 5000 Hz | |
| Mean flight altitude | 3767 m | |

Figure 2 shows the airborne experiment area around the South China Sea. The flight path of the aircraft is shown as the red line, and the total length of the flight path is about

115 km. This area is considered ideal for airborne altimetry test, since the bathymetry is shallow, therefore sea level anomaly related to ocean dynamics is unlikely to happen in this area, the SSH measured by altimeter should coincide well with the mean sea surface (MSS) of this region, as shown by the colored contours in Figure 2. Here we use the MSS2018 model from Technical University of Denmark (DTU), which is averaged over years of SSH measurements by satellite altimeters, so it can reflect the changes of SSH to a certain extent [18].

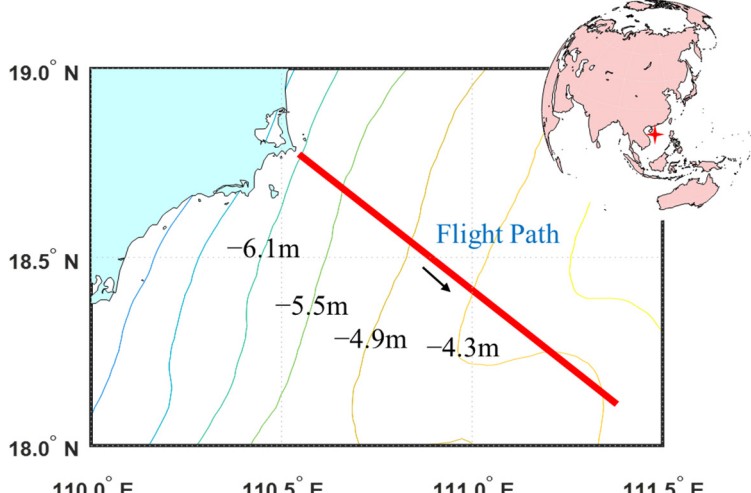

**Figure 2.** Map of the airborne experiment area. The red line represents the flight path, and the other colored curves represent the MSS contours.

### 2.2. SSH Determination Principle of SAR Altimeter

The altimeter is mainly used to measure the height of the sea surface relative to the reference ellipsoid, namely sea surface height (SSH). Figure 3 is the geometric schematic diagram of height measurement by radar altimeter. For the observed sea area, when the altimeter locates at point B, the SSH can be calculated as:

$$SSH = H - h \tag{1}$$

where $H$ is the flight height of the aircraft, $h$ is the distance between the altimeter and the sea surface, which is determined by timing half of the round-trip delay of the radar echo.

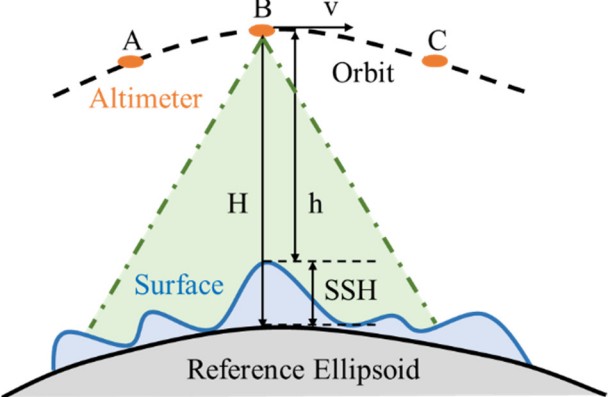

**Figure 3.** Geometric schematic diagram of height measurement by radar altimeter.

Therefore, the measurement precision of SSH depends on the orbit determination precision on the one hand, and the altimeter ranging precision on the other hand. Besides, multiple altimetry errors, such as systematic internal delay, ionospheric and tropospheric delay, ocean tides, etc., should be corrected first before SSH retrieval. However, for the airborne experiment of a confined space and time interval, those errors can be considered

as a bias and is removed by comparing with the EGM2008 (the details are discussed in Section 3.2). Therefore, the main errors of the altimeter that are discussed in this paper are the speckle noise and the altitude error of the aircraft. The speckle noise usually deteriorates the radar echo, to get the altimeter ranging result accurately, the waveform re-tracking method is used to for SSH retrieval.

The altitude of the aircraft often changes significantly due to atmosphere disturbance, in the airborne experiment, the altitude can vary by several meters along the flight, as shown in Figure 4. Yet the precision requirement of SSH is on centimeter-level, owing to centimeter-level precision of the orbit determination from GNSS equipment, so the raw echoes are height compensated first, then the echoes are de-ramped to achieve pulse compression.

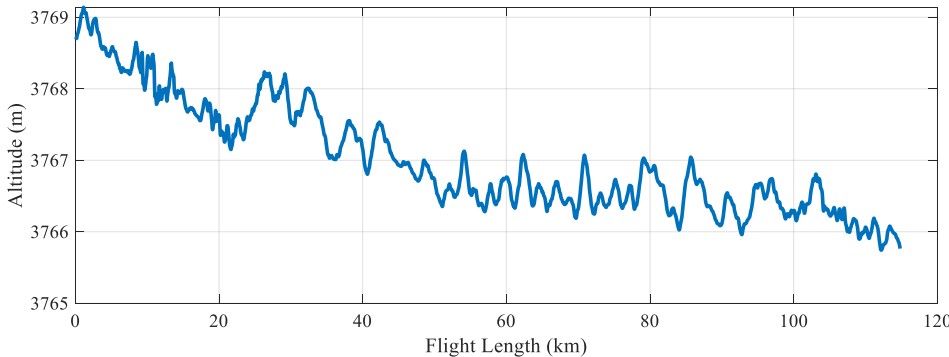

**Figure 4.** The variation of altitude along the flight path.

Affected by the speckle noise from surface waves, the SCR of one single echo is very low, leads to deformed waveform that can hardly be used for re-tracking. So multi-look processing (or multi-looking) is required to increase the SCR first. Due to the coherence between adjacent echoes, which is guaranteed by a high pulse repetition frequency (PRF) of 5000 Hz, a certain number of echoes can be grouped into a burst for synthetic aperture processing before multi-looking, during which the SCR is greatly improved by range migration correction (RMC) through phase compensation in the Doppler domain.

For echo of Doppler frequency $f_d$, the range correction is expressed as:

$$\Delta r(f_d) = \eta \frac{\lambda^2 H f_d^2}{8v^2} \qquad (2)$$

Let the range frequency be $f_r$, then the phase to be compensated is:

$$\Delta\Phi(f_d) = \exp(j2\pi f_r \cdot \frac{2\Delta r}{c}) = \exp(j\frac{2\pi f_r}{c} \cdot \frac{\eta\lambda^2 H f_d^2}{4v^2}) \qquad (3)$$

where $\eta$ is the earth curvature, $\lambda$ is the radar wavelength, $v$ is the aircraft speed, $c$ is the light speed.

After performing SAR processing and multi-looking, the echo model is established based on the airborne experiment parameters before waveform re-tracking. As the attitude of the aircraft changes drastically, the mis-pointing angle of antenna beam is considered for the establishment of the theoretical echo model by taking the variation of roll and pitch angle into account. Through least squares re-tracking, the epoch of an altimeter echo (the half-power point of the rising edge of the echo, unit: second) which is one of the parameters in the theoretical model can be get, then the SSH can be calculated based on Equation (1) afterwards. The flowchart of the airborne data processing is shown in Figure 5.

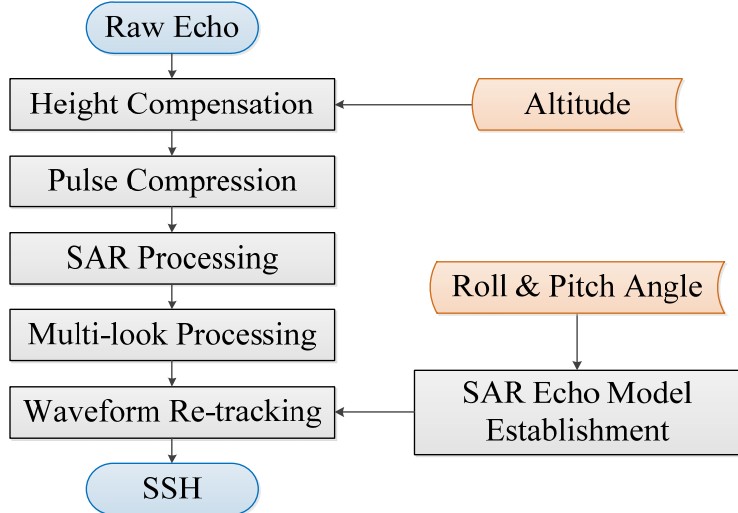

**Figure 5.** Flowchart of the airborne data processing procedure.

*2.3. Echo Model of Airborne SAR Altimeter*

In 1977, Brown proposed an explicit expression of the sea surface backscatter echo model [19]:

$$P(t) = P_{FS}(t) * q_s(t) * s_r(t) \tag{4}$$

where $P(t)$ is the average power of the received echo, $P_{FS}(t)$ is the impulse response function of the flat sea surface, $q_s(t)$ is the probability density function of surface wave height, and $s_r(t)$ is the point target response of radar system.

The SAR echo model is established based on the Brown model, in which:

$$q_s(t) = \frac{1}{\sqrt{2\pi}\sigma_s} \exp(-\frac{t^2}{2\sigma_s^2}) \tag{5}$$

$$s_r(t) = \frac{1}{\sqrt{2\pi}\sigma_p} \exp(-\frac{t^2}{2\sigma_p^2}) \tag{6}$$

where $\sigma_s = \frac{SWH}{2c}$, SWH is the significant wave height, $\sigma_p = 1.125/B$, $B$ is the bandwidth of transmitting signal.

Let $\sigma_c = \sqrt{\sigma_p^2 + \sigma_s^2}$, then the convolution of the above two functions is:

$$Y(t) = \frac{1}{\sqrt{2\pi}\sigma_c} \exp(-\frac{t^2}{2\sigma_c^2}) \tag{7}$$

Due to the synthetic aperture processing of the echoes in a burst, the radar beam has been sharpened, usually referred as Doppler beam sharping (DBS), and the real aperture beam is divided into several along-track sub-beams [20]. Each sub-beam corresponds to a Doppler strip along the flight track, and the strip width determines the azimuth resolution of SAR altimeter. Due to the relative velocity variation from the aircraft to the along-track strips, the echoes from different strips have different Doppler frequencies, and they are called sub-look echoes. After RMC of the sub-look echoes, the multi-look echo can be obtained [21,22].

Through the above analysis, it can be concluded that the idea of establishing the SAR echo model is as follows: firstly establishing single-look $P_{FS}(t)$ for each Doppler strip according to the Doppler frequencies, then convolving $P_{FS}(t)$ with delayed $Y(t)$, finally performing multi-look processing on the sub-look echoes [23,24].

To determine the impulse response function of a flat sea surface is the most challenging part. Since the geometric model of SAR altimeter usually involves convolution operations,

most of the existing models ignore the mis-pointing angle of antenna beam [25–28]. However, in airborne experiment, the roll angle and pitch angle both change drastically, as shown in Figure 6, which directly affect the echo waveform. If this is not considered during echo model establishment, significant tracking error will be brought in. Moreover, unlike the traditional round beam, the beam of this airborne experiment is elliptical, so the antenna pattern is also different from ordinary altimeter.

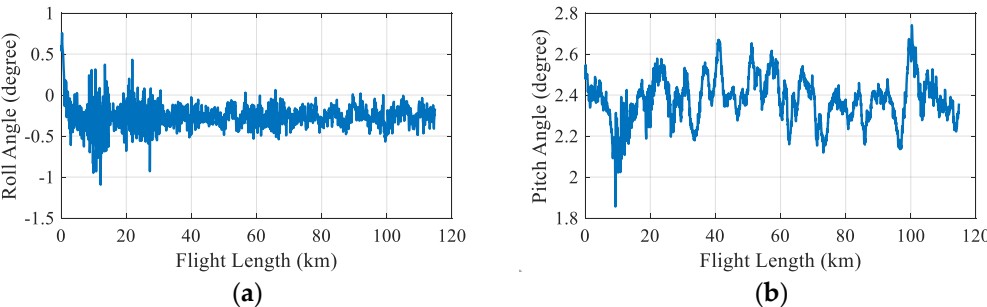

**Figure 6.** The variation of (**a**) roll angle and (**b**) pitch angle along the flight path.

To sum up, a SAR echo model that takes both mis-pointing angle and elliptical beam into account is needed. Reference [29] provides a SAR interferometric (SARIn) altimeter echo model which meets those requirements. The model in [29] is proposed to process the CrySat-2 SARIn power waveform, which could be affected by interferometer angle composed of mis-pointing angle and surface across-track slope. The CrySat-2 altimeter that operates on SARIn mode, is equipped with two antennas to form an interferometric baseline to measure the slope near the glacier margin. However, the airborne altimeter works in non-interferometric, nadir-looking mode, so the SARIn mode is beyond the scope of this paper. Therefore, by setting the baseline length of Equation (1) in [29] to zero, an echo model that applies to nadir SAR altimeter of elliptical beam footprint is obtained.

The impulse response function of the i-th sub-beam is expressed as:

$$P_{Fs}^i(t) = \frac{\lambda^2 G_0^2 D_0 c \sigma^0}{32\pi^2 H^3 \eta} \cdot J(t + \frac{\eta H \xi_k^2}{c}) \cdot \int_0^{2\pi} F(\rho_k \cos\vartheta - \xi_k) \cdot G(\rho_k, \vartheta) d\vartheta \tag{8}$$

where $G_0$ is the antenna gain, $D_0$ is the composite beam gain, $\sigma^0$ is the backscatter coefficient, $J(\cdot)$ is the step function, $\xi_k$ is the look angle, $F(\cdot)$ is the weighting coefficient, $\rho_k = \sqrt{\frac{ct}{\eta H} + \xi_k^2}$ accounts for the RMC, and $G(\rho_k, \vartheta)$ is the antenna pattern expressed as:

$$G(\rho_k, \vartheta) = \exp[-\frac{2(\rho_k \cos\vartheta - \mu)^2}{\gamma_1^2}] \cdot \exp[-\frac{2(\rho_k \sin\vartheta - \chi)^2}{\gamma_2^2}] \tag{9}$$

where $\mu$ is the pitch angle, $\chi$ is the roll angle, $\gamma_1$ and $\gamma_2$ are the 3 dB beam width along and across track, respectively.

Let $\tau$ be the echo epoch, then the single-look echo model is:

$$P_i(t) = Y(t - \tau) * P_{Fs}^i(t) \tag{10}$$

If the total number of looks is $M$, then the multi-look echo model is:

$$P(t) = \sum_{i=1}^{M} P_i(t) \tag{11}$$

When re-tracking the airborne SAR waveform, the mis-pointing angles (i.e., roll and pitch angle) recorded by on-board inertial measurement unit (IMU) are used as inputs, so the fitness between the theoretical echo model and the actual airborne waveform is

enhanced. This reduces the possibility of tracking failure caused by the low similarity between theoretical model and actual waveform, which improves the precision of SSH retrieval in return [30–32].

When other system parameters are determined, the SAR echo model can be expressed as a function of three parameters: echo epoch, SWH, and echo amplitude. Figure 7 shows the SAR echo model of different echo epoch and SWH.

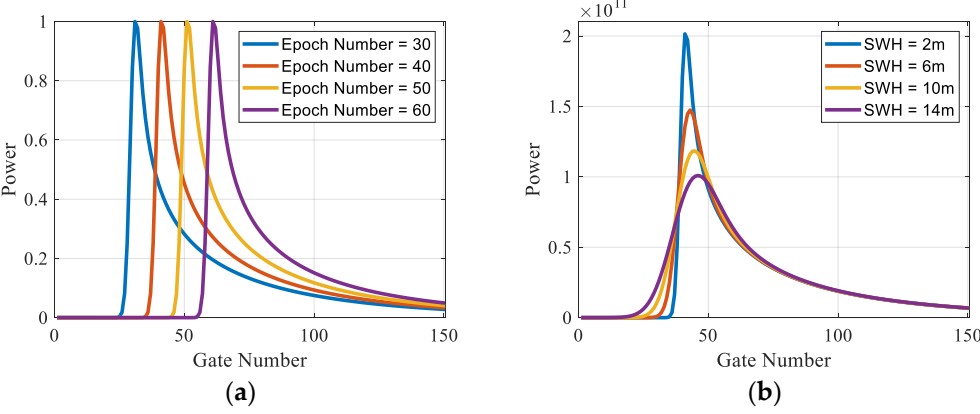

**Figure 7.** The SAR echo model with different (**a**) echo epoch (SWH = 2 m) and (**b**) SWH (Epoch = 40).

### 2.4. Re-Tracking Algorithm and Theoretical Precision

Re-tracking uses a certain algorithm to fit the echo model to the actual waveform to achieve the most consistent state, thus to retrieve parameters contained in the echo. In practical applications, by re-tracking the altimeter waveform, the SSH, SWH, and backscattering coefficient are obtained, which can be further used by oceanographers.

In this paper, the least squares algorithm (LSA) is utilized for waveform re-tracking. An iterative process is used to reach the optimal solution in the specific implementation [33–35]. The flowchart of re-tracking step is shown in Figure 8, in which the SAR echo model is abbreviated as an expression of three estimated parameters:

$$P = f(\tau, \text{SWH}, P_u) \tag{12}$$

where $P_u$ is the echo amplitude.

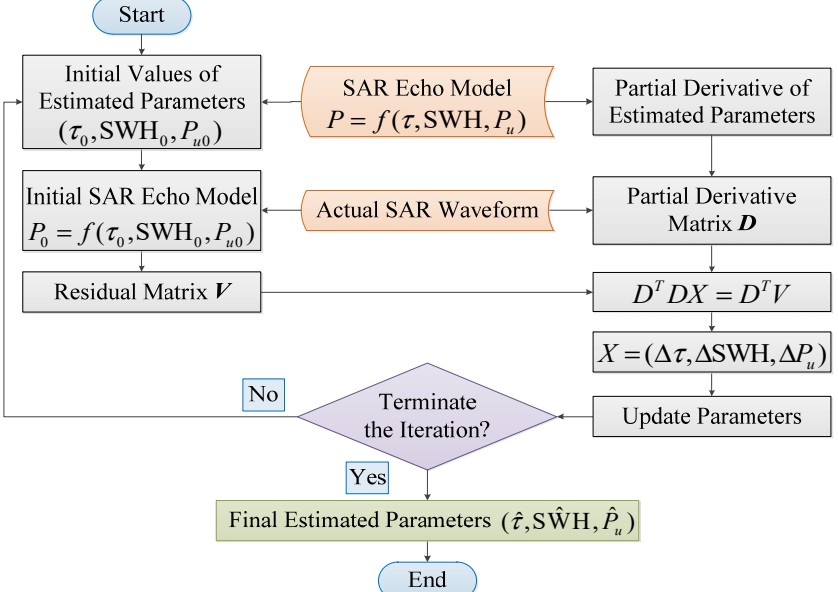

**Figure 8.** Flowchart of re-tracking algorithm implementation step.

By giving a set of initial values of the parameters to be estimated, the initial echo model and Taylor expansion are obtained, and the echo model is linearized. The basic idea of LSA is to minimize the square sum errors (SSE) between the linearized echo model and the actual airborne waveform. Let the partial derivative of SSE with respect to each parameter to be zero, and a matrix formula can be expressed as:

$$D^T D X = D^T V \tag{13}$$

where $D$ is the partial derivative matrix of the echo model; $V$ is the residual matrix; and $X$ is the incremental matrix of the estimated parameters, which is calculated to update the parameters.

Repeat the above steps until the SSE reach to a threshold that determines whether the echo model is close enough to the actual waveform, then stop the iteration process, and the parameter retrieval results are obtained.

It can be known from Equation (13) that the theoretical re-tracking precision of the estimated parameters depends on the partial derivatives of the model and the statistical characteristics of the waveform [36,37]. Transform Equation (13) as follows:

$$X = \left(D^T D\right)^{-1} D^T V = QV \tag{14}$$

If the total number of waveform sampling points is $N$, then the expressions of the estimated parameters $x_j$ and their variances $\sigma^2_{x_j}$ are:

$$x_j = q_{j1} v_1 + q_{j2} v_2 + \ldots + q_{jN} v_N \tag{15}$$

$$\sigma^2_{x_j} = q^2_{j1} \sigma^2_{v_1} + q^2_{j2} \sigma^2_{v_2} + \ldots + q^2_{jN} \sigma^2_{v_N} \tag{16}$$

where $q_{jn}$ is element of matrix $Q$, $v_n$ is element of matrix $V$, $\sigma^2_{v_n}$ is variance of $v_n$.

The corresponding standard deviations are the theoretical re-tracking precisions of a single SAR waveform. For the airborne radar altimeter, 1 Hz theoretical re-tracking precision is commonly used since the attitude of aircraft changes drastically. Assuming the theoretical re-tracking precision of a single SAR waveform is $\sigma$, and the number of SAR waveforms accumulated in 1 s interval is $N$, then the 1 Hz theoretical re-tracking precision is calculated as:

$$\Psi = \sigma / \sqrt{N} \tag{17}$$

## 3. Results

### 3.1. Waveform Re-Tracking and SSH Retrieval Results

During the processing of airborne data, 256 consecutive echoes are grouped into a burst to perform SAR processing. For the received burst echoes, the first step is to carry out azimuth fast Fourier transform (FFT); then perform RMC by phase compensation in the range domain (burst echoes along azimuth is in the Doppler domain after FFT). Convert the burst echoes accomplished with RMC processing back into time domain, and superpose the echoes, a SAR echo is then obtained. To further improve the SCR, 20 SAR echoes are accumulated incoherently.

Figure 9a displays the Ku-band airborne waveforms before and after SAR processing. The blue waveform is generated by superposing 5120 de-ramped echoes; and the red one represents the waveform after SAR processing and multi-looking. It can be seen that the SCR has been greatly improved. The Ka-band has similar results, as shown in Figure 9b. Note that in order for clear comparison, the y-axis which represents the waveform power has been normalized.

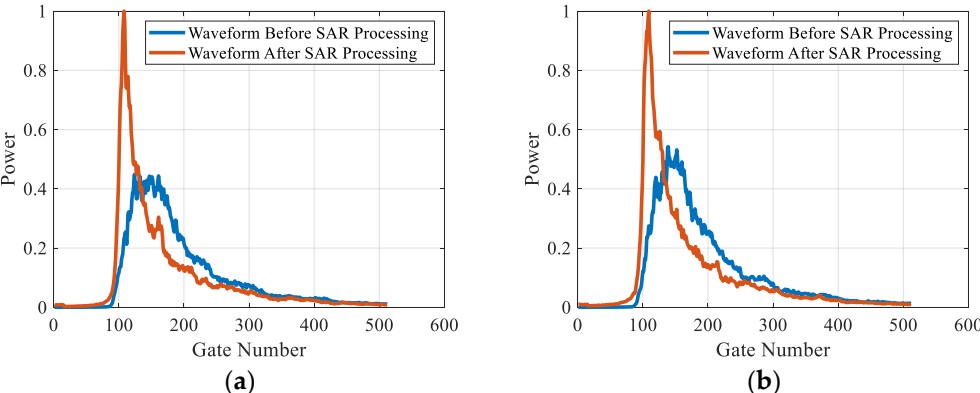

**Figure 9.** Airborne waveforms before and after SAR processing of (**a**) the Ku-band and (**b**) the Ka-band.

After SAR processing and multi-looking of all of the airborne data, 1332 echoes are obtained of Ku-band and Ka-band respectively, as shown in Figure 10, which correspond to an along-track flight length of about 115 km; the azimuth and range direction are indicated by yellow arrows.

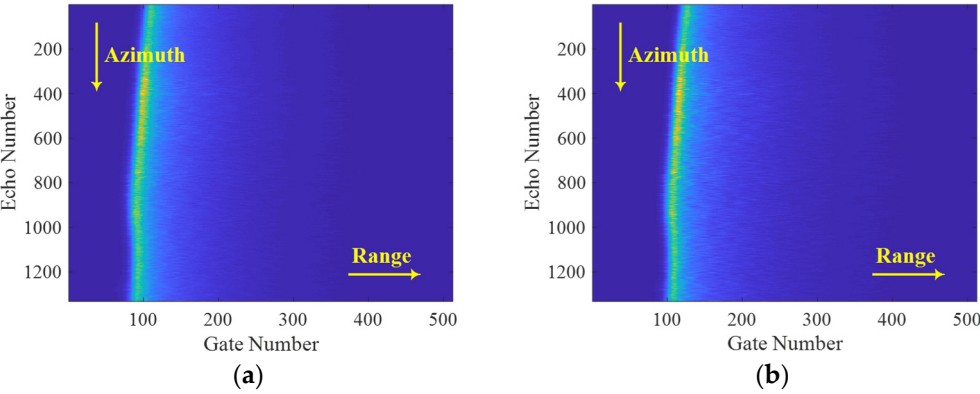

**Figure 10.** Airborne multi-look SAR echoes of (**a**) the Ku-band and (**b**) the Ka-band.

Figure 11a shows the re-tracking iteration of a Ku-band normalized airborne waveform. The blue curve is the normalized airborne waveform, the red curve is the initial echo model, and the yellow, purple, and green curves are the echo models after the 1st, 3rd, and 14th iterations, respectively. It can be seen that as the iteration number increases, the echo model gradually approaches to the airborne waveform. Figure 11b shows the corresponding re-tracking result, where the blue curve is the normalized airborne waveform and the red curve is the final echo model. It can be seen that the final echo model is in good agreement with airborne waveform. Similar steps are performed on Ka-band normalized airborne waveform, the re-tracking iteration and result are shown in Figure 11c,d.

The echo epoch is directly obtained by re-tracking, and SSH which is the distance from the instantaneous ocean surface to the reference ellipsoid can be retrieved according to Equation (1), by taking the initial sampling distance into account.

All of the airborne SAR waveforms in Ku and Ka band are re-tracked, the complete retrieved SSH is shown in Figure 12a, and a section of Figure 12a is enlarged and shown in Figure 12b. The abscissa of Figure 12 is the flight length, and the ordinate is the SSH in meters. The blue and red curve in Figure 12 are the SSH results of the Ku and Ka band, respectively.

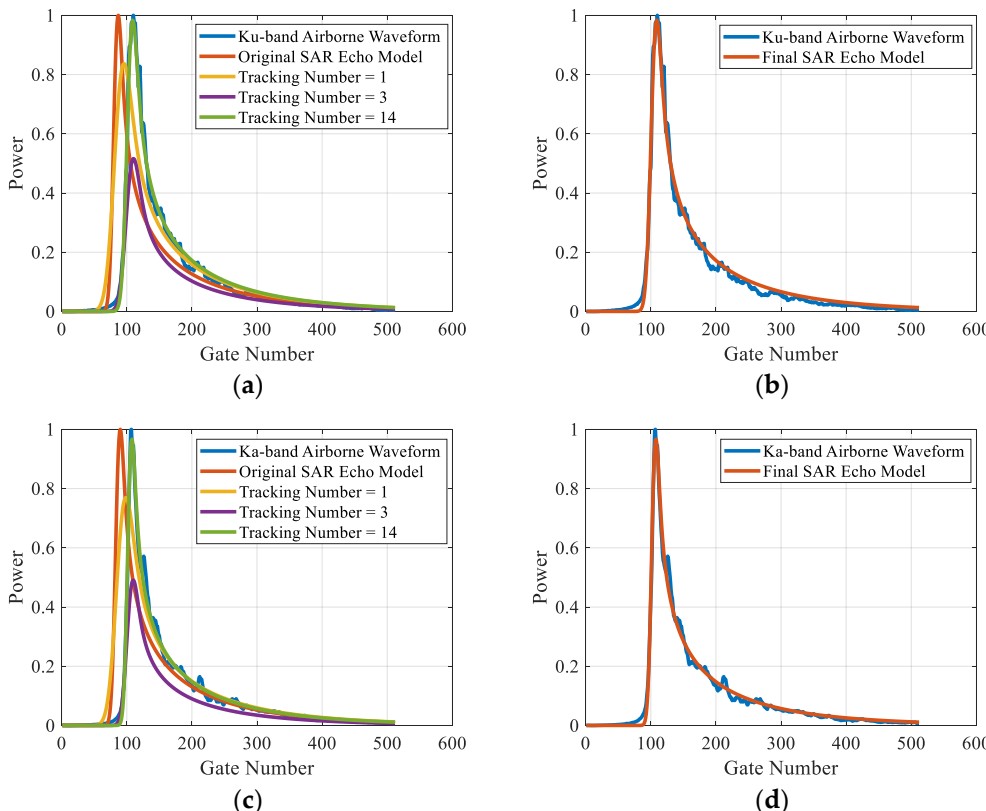

**Figure 11.** The waveform re-tracking iteration and result of Ku-band (**a**,**b**) and Ka-band (**c**,**d**).

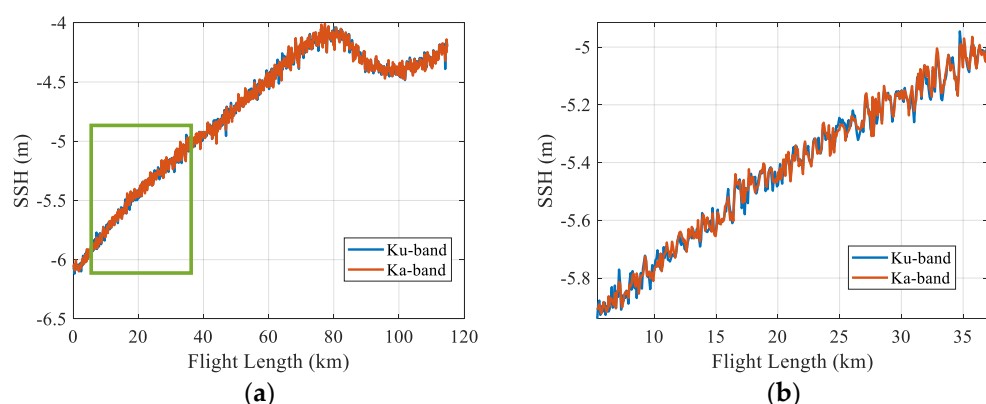

**Figure 12.** The SSH retrieval results of Ku-band (blue curve) and Ka-band (red curve). (**a**) The complete SSH retrieval results. (**b**) The enlarged section that is indicated in (**a**).

### 3.2. Verification and Comparison of the Ku/Ka Dual-Band SSH Data

The retrieved SSH data are verified by comparing to the public SSH data along the flight path at the experiment day [17]. The public SSH data are generated from the ocean general circulation model (OGCM) by assimilating multiple satellite altimeters and in situ observation dataset. The public SSH data are referenced to the geoid, while the retrieved SSH data measured by airborne altimeter are referenced to the WGS84 ellipsoid, so the public SSH data need to be transformed to the same reference. Here we use EGM2008 geoid model for this processing. Since the grid resolution of the public SSH is only $1/12°$, which is much coarser than that of the airborne results, the public SSH is interpolated to the same spatial resolution first for the comparison, and the result is shown in Figure 13. Note that the bias caused by instrument internal delay and atmosphere such as dry tropospheric refraction, wet tropospheric refraction and sea-state effects has been removed by subtracting a bias evaluated from the airborne SSH [38,39]. In Figure 13, the blue curve is the SSH

retrieved from Ku-band, the red curve is the SSH retrieved from Ka-band, and the green curve is the public SSH data. It can be seen that the variation trends of three SSH data sets are in good consistency, which proves the correctness of parameter retrieval method and results. The SSH from the start to the end of the flight route changes as much as 190 cm mainly due to the geoid variation since ocean dynamics such as eddies are not common near the experiment area.

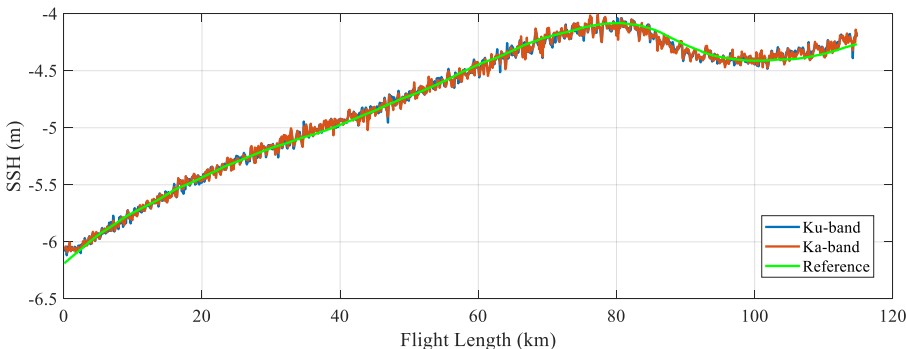

**Figure 13.** Verification of dual-band SSH retrieval results. The blue curve is the SSH data of Ku-band retrieval, the red curve is the SSH data of Ka-band retrieval, and the green curve is the public SSH data along the flight path at the experiment day.

In order to compare the retrieval precision of Ku-band and Ka-band, the original SSH retrieval results are interpolated into 1 Hz resolution (about 84 m along-track), and the root-mean-square errors (RMSE) are calculated as precision by taking the public SSH data as reference value. The 1 Hz RMSE results are 4.85 cm and 4.73 cm for Ku and Ka band, respectively, which suggests that the two bands have comparative accuracies, it seems that Ka-band has no obvious ranging advantage compared with Ku-band. To evaluate if the result is correct or not, the 1 Hz theoretical accuracies are also calculated.

As discussed in Section 2.2, the retrieval precision of SSH depends on both the orbit determination precision and the altimeter ranging precision. The method in Section 2.4 is used to calculate the theoretical precision of altimeter ranging, while the altitude determination precision deteriorates as the aircraft moves away from the GNSS base station [40,41]. In this airborne experiment, the base station is located near the coast. When the plane is within 30 km radius of the GNSS base station, the 1 Hz orbit determination precision is about 3 cm; when the plane is more than 50 km away from base station, the 1 Hz precision decreases to about 5~10 cm. After assessing the data quality along the entire flight path, it is estimated that the overall 1 Hz precision of altitude determination is about 4~5 cm. The measurements of altimeter ranging and altitude determination can be regarded as two independent processes, and therefore the final precision of the retrieved SSH is:

$$\sigma_{\text{SSH}} = \sqrt{\sigma_{rg}^2 + \sigma_{alt}^2} \tag{18}$$

where $\sigma_{rg}$ is the precision of altimeter ranging, which is calculated separately for Ku and Ka band according to Equation (17) in Section 2.4, and $\sigma_{alt}$ is the precision of altitude determination, which is the same for both frequencies.

Based on Equation (18) and method in Section 2.4, the 1 Hz theoretical precision of dual-band airborne data with or without considering altitude determination error is calculated and shown in Table 2. 1 m $\leq$ SWH $\leq$ 4 m are considered for most of the scenario. It can be found that the theoretical precisions of Ku-band and Ka-band are indeed comparative, and the actual measurement precisions of 4.85 cm and 4.73 cm are both reasonable.

**Table 2.** 1 Hz theoretical height measurement precision of airborne data.

| SWH | Ku-Band | Ka-Band | Ku-Band | Ka-Band |
|---|---|---|---|---|
| | (Without Considering Altitude Determination Error *) | | (With Considering Altitude Determination Error *) | |
| 1 m | 1.12 cm | 1.07 cm | 4.15~5.12 cm | 4.14~5.11 cm |
| 2 m | 1.43 cm | 1.37 cm | 4.25~5.20 cm | 4.23~5.19 cm |
| 3 m | 1.70 cm | 1.63 cm | 4.35~5.28 cm | 4.32~5.26 cm |
| 4 m | 1.93 cm | 1.84 cm | 4.44~5.36 cm | 4.40~5.33 cm |

* The altitude determination error is about 4~5 cm.

### 3.3. Analysis of Re-Tracking Precision from Re-Processed Airborne Data

The ALtika altimeter on SARAL satellite has a bandwidth of 480 MHz while other Ku-band satellite altimeters only have 320 MHz bandwidth. The reason for that is Ka-band can adopt larger bandwidth due to its higher central frequency. However, in this airborne experiment, the two band used exactly the same bandwidth of 900 MHz. So we wonder if the SSH measurement precision is affected by transmitting bandwidth. To verify this assumption, we first calculate the 1 Hz theoretical height measurement precision under different bandwidth, with the rest of system parameters remaining the same, the results are shown in Table 3. It can be seen that the precision does get better as the bandwidth increases for both Ku and Ka band. It is easy to understand since a larger bandwidth leads to higher ranging precision, but worse signal-to-noise ratio (SNR) at the same time. However, the increased thermal noise has minor effect on the altimeter system since the original SNR is high enough to omit the impact [42]. Moreover, by analyzing the theoretical re-tracking precision method thoroughly, it is found that the mathematic reason for this result is transmitting bandwidth affects the echo model, and echo model determines the theoretical precision. The basis for this conclusion can be deduced from Equations (6), (7), (10), (14) and (16).

**Table 3.** 1 Hz theoretical height measurement precision under different bandwidth setting.

| Bandwidth | Ku-Band | Ka-Band |
|---|---|---|
| 300 MHz | 2.94 cm | 2.82 cm |
| 500 MHz | 2.05 cm | 1.96 cm |
| 700 MHz | 1.66 cm | 1.59 cm |
| 900 MHz | 1.43 cm | 1.37 cm |

SWH = 2 m (1 sigma of the global sea condition).

To make the airborne experiment more similar to the actual satellite design, the Ku-band and Ka-band raw de-ramped echoes are re-processed to achieve bandwidths of 320 MHz and 480 MHz, respectively. The realization is performed by low-pass filtering in the range-frequency domain. After re-processing the dual-band airborne data according to Figure 5, the new SSH retrieval results are obtained and shown in Figure 14. The statistical 1 Hz RMSE of Ku and Ka band are 5.41 cm and 4.99 cm, and the precision ratio is 1.08. Since the airborne experiment is mainly used for qualitative comparison, the precision ratio of Ku-band and Ka-band SSH retrieval results is also calculated to evaluate the precision improvement. The above results indicates that Ka-band has a slight advantage compared with Ku-band.

Under the new bandwidth setting (Ku: 320 MHz, Ka: 480 MHz), the 1 Hz theoretical height measurement precision of dual-band airborne data are re-calculated and shown in Table 4. It can be found that the theoretical precision of Ka-band is better than Ku-band, and the precision rates with or without considering altitude determination error are approximately 1.1 and 1.4, respectively. The processed precision ratio given before is 1.08, which meets the theoretical expectation.

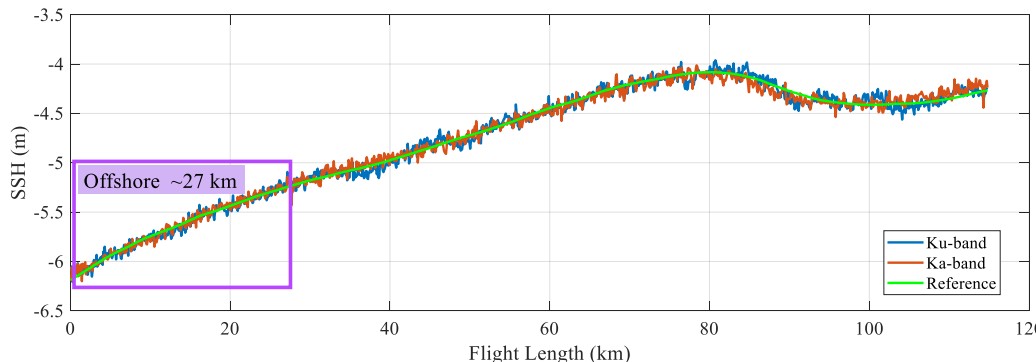

**Figure 14.** Comparison of re-processed dual-band SSH retrieval results. The blue curve is the SSH retrieval result of Ku-band under 320 MHz bandwidth, the red curve is the SSH retrieval result of Ka-band under 480 MHz bandwidth, and the green curve is the public SSH data of the flight path.

**Table 4.** 1 Hz theoretical height measurement precision of re-processed airborne data.

| SWH | Ku-Band | Ka-Band | Ku-Band | Ka-Band |
|---|---|---|---|---|
| | (Without Considering Altitude Determination Error *) | | (With Considering Altitude Determination Error *) | |
| 1 m | 2.53 cm | 1.71 cm | 4.73~5.60 cm | 4.35~5.28 cm |
| 2 m | 2.80 cm | 2.01 cm | 4.88~5.73 cm | 4.48~5.39 cm |
| 3 m | 3.12 cm | 2.32 cm | 5.07~5.89 cm | 4.62~5.51 cm |
| 4 m | 3.42 cm | 2.59 cm | 5.26~6.05 cm | 4.76~5.63 cm |
| Ratio | 1.32~1.48 | | 1.06~1.11 | |

* The altitude determination error is about 4~5 cm.

## 4. Discussion

In Section 3.3, after re-processing the dual-band airborne data into new bandwidth setting (Ku: 320 MHz, Ka: 480 MHz), the Ku/Ka precision ratio is obtained as 1.08, which indicates that Ka-band has a slight advantage compared with Ku-band. To further explore the potential superiority of Ka-band, we re-analyzed a subset of SSH retrieval data which is acquired within the 27 km range offshore, as marked by purple box in Figure 14. This dataset has caught our attention since its oscillation is rather small in comparison with the rest of SSH retrieval results. The reason for that is the altitude determination precision deteriorates as the aircraft moves away from the GNSS base station.

As discussed in Section 3.2, the 1 Hz altitude determination precision is about 3 cm when the plane is within 30 km radius from the base station. After removing the 3 cm altitude determination error according to Equation (18), the SSH retrieval precisions are 3.31 cm and 2.42 cm for Ku and Ka band, respectively, and the precision ratio is 1.37, which matches the ratio shown in Table 4. This result demonstrates that the Ku/Ka precision ratio is possible to achieve 1.4 within the 27 km radius of offshore area, which further proves that the Ka-band has better performance.

Based on the above discussion, it can be concluded that since Ka-band can adopt larger bandwidth thanks to higher radar frequency than that of Ku-band. Therefore, the Ka-band is more promising in achieving higher range precision by taking into account the different expected bandwidths on satellite-borne altimeters (Ku: 320 MHz, Ka: 480 MHz). With SARAL already providing high-quality altimetry data, it is indicated that the Ka-band SAR altimeter will show better performance benefiting from its higher resolution and range precision.

In addition to bandwidth, the SSH retrieval precision is also affected by beam coverage, which can be demonstrated by calculating the theoretical height measurement precision under different 3 dB beam width along-track and across-track. The result can be simply described as the fact that a larger beam width leads to better precision for both the Ku-band and Ka-band. While the airborne experiment discussed in this paper has a narrow beam

coverage known from Section 2.1, which limits the height measurement precision to some extent. In 2022, there will be another more in-depth airborne experiment which will adopt round antennas. The 3 dB beam width of Ku and Ka-band will be approximately 25° and 15°, respectively. With wider beam, the SSH measurement precision is expected to gain better performance.

## 5. Conclusions

Benefiting from higher ranging precision and simpler system design, the Ka-band SAR altimeter is attracting growing attention. To sufficiently study the SAR altimeter data processing method and explore the system advantage of the Ka-band, a Ku/Ka dual-band altimeter airborne experiment was carried out over the South China Sea on 6 November 2021. In this paper, a series of treatments are performed on dual-band airborne data, and the SSH is retrieved successfully and analyzed thoroughly. The main conclusions are as follows:

(1) This airborne experiment has acquired the Ku/Ka dual-band echo data simultaneously for the first time, and thus the comparison results are profound.

(2) The dual-band retrieval SSH data are compared with the public SSH data of the flight path for the experiment day, and the change trends of three SSH data sets are very consistent, which proves the correctness of SSH retrieval method and results.

(3) By calculating the theoretical precision of waveform re-tracking and re-processing the dual-band airborne data into different bandwidths in reference to those chosen by satellite-borne altimeters, it is demonstrated that the Ku/Ka precision ratio is possible to achieve 1.4 within the 27 km offshore area, which indicates that Ka-band has better performance than that of Ku-band.

**Author Contributions:** Conceptualization, X.L. and H.S.; methodology, X.L. and W.K.; software, X.L.; validation, X.L., W.K. and H.S.; formal analysis, X.L.; investigation, X.L. and W.K.; resources, H.S. and Y.L.; data curation, W.K.; writing—original draft preparation, X.L.; writing—review and editing, W.K.; visualization, X.L.; supervision, H.S. and Y.L.; project administration, H.S. and Y.L.; funding acquisition, H.S. and Y.L. All authors have read and agreed to the published version of the manuscript.

**Funding:** This research was funded by "WenHai" Project Fund of Shandong Province for Pilot National Laboratory for Marine Science and Technology (Qingdao), grant number 2021WHZZB1600.

**Data Availability Statement:** Not applicable.

**Acknowledgments:** The authors would like to acknowledge the support of everyone who assisted with the airborne experiment involved in this study.

**Conflicts of Interest:** The authors declare no conflict of interest.

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
