# Peer review of "Performance Analysis of Ku/Ka Dual-Band SAR Altimeter from an Airborne Experiment over South China Sea"

_remotesensing, doi:10.3390/rs14102362_

Round 1

Reviewer 1 Report

  1. This paper analyzes the performance of Ku/Ka dual-band SAR altimeter based on an airborne experiment. The paper provides good reference information for further studies and for practical applications in this field.
  2. The paper is generally written well but does not meet the publication standard. A proofreading is recommended to correct grammatical errors in the paper. Some examples are given here for authors’ reference: in Line 93, “The results indicates…” should be “indicate…”; in Line 120, “it is no larger than…” should be “not larger than…”; in Line 237, “scope of this pater” should be “paper”, etc.
  3. The term “accuracy” used in the paper is actually “precision”. To evaluate accuracy, the true value should be known. Please check all the contents related to “accuracy” and make correction accordingly.

Reviewer 2 Report

This paper studies the performance of Ku/Ka-band dual-frequency SAR altimeter based on an airborne experiment carried out over South China Sea on November 6, 2021. The experiment data were processed to estimate the SSH retrieval accuracy after procedures such as height compensation, range mitigation correction, multi-look processing, and waveform re-tracking. The retrieved SSH for both frequencies showed good agreement with the public SSH data. The accuracy of Ka-band had better performance than Ku-band.

This article is very useful for the community of this research field. The manuscript is written clearly, but the discussion section (Section 4) is insufficient and there are some typos. In addition, if there are any subjects for the development of satellite-borne Ka-band SAR altimeter found from airborne experiment, they should be pointed out. So this paper may be published after minor revision.

Specific Comments

L178: range frequency --> radar frequency (?)

L237: this pater --> this paper (?)

L398: find --> found

Reviewer 3 Report

See attached file.

Reviewer 4 Report

General Comments

I recommend that this paper is accepted for publication with minor  revisions

It presents a new experimental results from an airborne Ka and Ku band SAR altimeter.

However, the experimental results do not show a clear difference between the accuracy of the Ka band and Ku band altimeters as is implied  in the conclusions, in the introduction section and in the abstract. This is calculated as theoretically possible by taking into account the different expected bandwidths on satellite borne Ka and Ku altimeters and this should  be made clear.

Figure 15 is only discussed very briefly in the text. It should be discussed in more detail to justify its inclusion (or not included)

Round 2

Reviewer 3 Report

I recommend the manuscript to be accepted for publication as a note, which reports a progress in the research field.